# Anti-Inflammatory, Barrier Maintenance, and Gut Microbiome Modulation Effects of *Saccharomyces cerevisiae* QHNLD8L1 on DSS-Induced Ulcerative Colitis in Mice

**DOI:** 10.3390/ijms24076721

**Published:** 2023-04-04

**Authors:** Qianjue Hu, Leilei Yu, Qixiao Zhai, Jianxin Zhao, Fengwei Tian

**Affiliations:** 1State Key Laboratory of Food Science and Technology, Jiangnan University, Wuxi 214122, China; 2School of Food Science and Technology, Jiangnan University, Wuxi 214122, China; 3National Engineering Research Center for Functional Food, Jiangnan University, Wuxi 214122, China

**Keywords:** probiotics, *S. cerevisiae* QHNLD8L1, ulcerative colitis, intestinal barrier, gut microbiota

## Abstract

The use of probiotics has been considered as a new therapy option for ulcerative colitis (UC), and yeast has recently received widespread recommendation for human health. In this study, the probiotic characteristics of four yeast strains, *Saccharomyces boulardii* CNCMI-745, *Kluyveromyces marxianus* QHBYC4L2, *Saccharomyces cerevisiae* QHNLD8L1, and *Debaryomyces hansenii* QSCLS6L3, were evaluated in vitro; their ability to ameliorate dextran sulfate sodium (DSS)-induced colitis was investigated. Among these, *S. cerevisiae* QHNLD8L1 protected against colitis, which was reflected by increased body weight, colon length, histological injury relief, decreased gut inflammation markers, and intestinal barrier restoration. The abundance of the pathogenic bacteria *Escherichia–Shigella* and *Enterococcaceae* in mice with colitis decreased after *S. cerevisiae* QHNLD8L1 treatment. Moreover, *S. cerevisiae* QHNLD8L1 enriched beneficial bacteria *Lactobacillus*, *Faecalibaculum,* and *Butyricimonas,* enhanced carbon metabolism and fatty acid biosynthesis function, and increased short chain fatty acid (SCFAs) production. Taken together, our results indicate the great potential of *S. cerevisiae* QHNLD8L1 supplementation for the prevention and alleviation of UC.

## 1. Introduction

Inflammatory bowel disease (IBD), including ulcerative colitis (UC) and Crohn’s disease (CD), affects 6–8 million individuals worldwide [1]. UC is a recurrent and remitting mucosa inflammation that begins distally and can spread proximally to involve the whole colon [2]. The clinical symptoms of UC include bloody diarrhea, abdominal pain, and fecal incontinence. Disordered microbiota may lead to the onset of UC by breaking the intestinal epithelial barrier and infiltrating inflammatory cytokines [3]. Once the intestinal barrier is compromised, increased intestinal permeability accelerates UC progression, and tight junction proteins (TJPs) are crucial for maintaining intestinal epithelial permeability. Inflammatory responses and damage mainly result from the release of pro-inflammatory cytokines by macrophages [4]. The use of 5-aminosalicylic acid, corticosteroids, and biologics are traditional therapeutic options for UC [5]; however, there is a pressing need to develop new and more efficient alternatives owing to the low effectiveness and significant adverse effects of current medications.

Probiotics contain live nonpathogenic microorganisms that meet the nutritional needs of the host and inhibit harmful microbial colonization when they are ingested in sufficient amounts [6]. Probiotics are used to treat diseases by modifying the microbial community, enhancing intestinal barrier integrity, and maintaining a balanced immune response [7]. Recent studies have shown that probiotics as biological agents can relieve UC [8,9]. However, research on UC treatment has primarily focused on bacteria. Unlike bacteria, certain yeasts are antibiotic resistant and do not share resistance genes with bacteria [10]. Yeasts may benefit the host by binding pathogens to the cell surface and producing accessible anti-inflammatory extracellular components [11], which meets the primary standards and prerequisites for probiotics [12]. Thus, yeast strains show great promise for the probiotic industry.

For decades, *Saccharomyces boulardii* has been the focus of most clinical and animal studies on yeast probiotics. As a model strain, *S. boulardii* CNCMI-745 has been utilized to prevent intestinal damage and inflammation in recent years [13]. Yeasts may possess potential probiotic characteristics; however, studies on their probiotic functions in human health are rare. Although several yeasts have shown promising effects in the treatment of colitis [14,15], it is still unknown whether other yeast strains have a therapeutic potential. Additionally, a few researchers have explored the mechanisms by which yeasts regulate the immune response in UC. Therefore, investigating the anti-inflammatory properties of yeasts is critical for the development of potential probiotics. This study aimed to further understand probiotic yeasts by examining the impact and action mechanism of several yeast strains on mice with dextran sulfate sodium (DSS)-induced colitis.

## 2. Results

### 2.1. In Vitro Probiotic Characteristics

We evaluated yeast viability under simulated stomach and small intestine conditions. Table 1 shows the survival rates of the different yeasts ranging from 76.92 to 82.93%. After treatment with simulated gastric fluid, S. boulardii CNCMI-745 exhibited the highest survival rate (89.08%), followed by Saccharomyces cerevisiae QHNLD8L1, Kluyveromyces marxianus QHBYC4L2, and Debaryomyces hansenii QSCLS6L3 (*p* < 0.05). The survival rates of these yeasts were moderately lower in the intestinal fluid after 4 h; S. cerevisiae QHNLD8L1 reached the highest rate at 82.93% (*p* < 0.05), suggesting that it was tolerant to the intestinal environment. The adhesion rates of the four yeast strains on HT-29 cells exceeded 10%, among which D. hansenii QSCLS6L3 had a relatively weaker adhesion ability than the other yeasts did (*p* < 0.05). Thus, these four yeast strains showed potential probiotic properties, including high gastrointestinal tolerance and adhesion capacity.

### 2.2. Effects of Yeast Strains on Tight Junction Protein Expression in Caco-2 Cells

To assay the modulation of DSS-induced responses by different yeasts, we added yeast strains (10^7^ CFU/mL) to the Caco-2 cells model co-cultured with 2.5% DSS for 12 h. In our study, changes in TJPs in DSS-induced Caco-2 cells were detected after yeast intervention (Figure 1). Claudin-1 and Occludin relative expression levels were downregulated by DSS and were reversed by *S. boulardii* CNCMI-745 and *S. cerevisiae* QHNLD8L1 (*p* < 0.05). Additionally, *S. boulardii* CNCMI-745 significantly increased ZO-1’s relative expression, *K. marxianus* QHBYC4L2 significantly increased Occludin and ZO-1’s relative expressions (*p* < 0.05), and *D. hansenii* QSCLS6L3 had some upregulatory effects on these genes, but they were not significant (*p* > 0.05). Thus, we speculate that these yeasts may improve the barrier function in intestinal epithelial cells.

### 2.3. Effects of Yeast Strains on Symptoms of Mice with Colitis

Different yeasts have varying effects on colitis in mice, as confirmed by the changes in body weight, colon length, and the disease activity index (DAI) score. The weight loss rate on day seven in DSS induced-mice was 17.87 ± 5.60%, and the administration of *S. boulardii* CNCMI-745 and *S. cerevisiae* QHNLD8L1 significantly reduced the colitis symptoms, which are characterized by weight loss rates of 9.35 ± 3.50% and 11.09 ± 5.86%, respectively (*p* < 0.05) (Figure 2A). The DAI score of the DSS group was 3.73 ± 0.38, while that of *S. boulardii* CNCMI-745 group was 2.84 ± 0.36, with a decrease of 23.86%. *K. marxianus* QHBYC4L2 and *S. cerevisiae* QHNLD8L1 had DAI scores that were significantly reduced by 28.62% and 39.41%, respectively (*p* < 0.05) (Figure 2B). Colon shortening marks the emergence of inflammation. The colon length in the DSS group was 4.37 ± 0.32 cm. *S. boulardii* CNCMI-745, *K. marxianus* QHBYC4L2, and *S. cerevisiae* QHNLD8L1 significantly recovered DSS-induced colon shortening––the colon lengths of mice in the three groups were 1.21, 1.16, and 1.28 times greater than that in the DSS group, respectively (*p* < 0.05). However, the difference between the *D. hansenii* QSCLS6L3 group and the DSS group was not significant (*p* > 0.05) (Figure 2C,D).

### 2.4. Effects of Yeast Strains on Colonic Histological Injury

Histological damage was determined using Hematoxylin and Eosin (H&E) staining. The colonic tissue of DSS-induced mice was extensively ulcerated with mucosal epithelium loss, with an absence of crypts and severely infiltrated inflammatory cells (Figure 3A). The histological score of the DSS group was 10.25 times greater that of the control group, and *S. cerevisiae* QHNLD8L1 markedly reduced the histological scores (*p* < 0.05), showing a relatively intact mucosal epithelium, partially disappeared crypts, and less inflammatory infiltration. The pathological injury and histological scores were also decreased in other yeasts, but the differences were not statistically significant (*p* > 0.05) (Figure 3B).

Mucin glycoproteins in the secretory granules of goblet cells and the mucous layer of colonic epithelial cells were stained with Alcian Blue (AB). The mucus layer was severely disrupted by DSS, and the goblet cells and mucins were greatly reduced in the DSS group, with almost no blue coloration (Figure 3C). However, the mucin injury was significantly improved by the yeast treatment, with the exception of *D. hansenii* QSCLS6L3 treatment. The number of goblet cells in the *S. cerevisiae* QHNLD8L1 group was much higher than that in the DSS group (*p* < 0.05) (Figure 3D), indicating that *S. cerevisiae* QHNLD8L1 boosted goblet cell maturation and expanded mucin secretion more effectively than the other yeasts did.

### 2.5. Effect of Yeast Strains on Inflammatory Cytokines and Enzymes in Colitis Mice

The concentrations of inflammatory cytokines in the colon were measured to evaluate the inflammatory modulation by different yeasts. The levels of TNF-α, IL-6, and IL-1β were increased, and that of IL-10 was decreased after DSS exposure in mice with colitis. Yeast administration, except for *D. hansenii* QSCLS6L, decreased the level of TNF-α significantly (*p* < 0.05) (Figure 4A). The level of IL-6 was reduced to some degree after yeast intervention; however, this reduction was not significant (*p* > 0.05) (Figure 4B). Additionally, IL-1β concentration in DSS-induced mice with colitis was significantly reduced by *S. cerevisiae* QHNLD8L1 (*p* < 0.05) (Figure 4C), and IL-10 concentrations were significantly increased by *S. boulardii* CNCMI-745 and *S. cerevisiae* QHNLD8L1 (*p* < 0.05) (Figure 4D). The levels of MPO in mice were upregulated nearly two-fold by DSS, which was reversed by *S. boulardii* CNCMI-745 and *S. cerevisiae* QHNLD8L1 (*p* < 0.05) (Figure 4E). In addition to *D. hansenii* QSCLS6L, yeast supplementation significantly downregulated the iNOS concentration (*p* < 0.05) (Figure 4F). COX-2 concentrations in mice with colitis increased significantly (*p* < 0.05), and the yeast treatment had no significant negative effect (*p* > 0.05) (Figure 4G). Overall, *S. cerevisiae* QHNLD8L1 showed a better inhibition ability than the other yeast strains did in terms of inflammatory regulation.

### 2.6. Effect of Yeast Strains on Tight Junction Protein in Colitis Mice

After DSS exposure, the expression levels of Claudin-1, Occludin, and ZO-1 were decreased significantly, and so were their concentrations, except for ZO-1 (*p* < 0.05). Both the concentration and expression levels of Claudin-1 increased after *S. cerevisiae* QHNLD8L1 treatment in mice with colitis (*p* < 0.05) (Figure 5A,D). Occludin concentration did not differ significantly between the DSS and tested strain groups (*p* > 0.05). However, the expression levels of Occludin in the *K. marxianus* QHBYC4L2 and *S. cerevisiae* QHNLD8L1 groups increased significantly (*p* < 0.05) (Figure 5B,E). Additionally, *S. boulardii* CNCMI-745 and *S. cerevisiae* QHNLD8L1 increased the expression levels of ZO-1 in DSS-induced mice, and *S. cerevisiae* QHNLD8L1 had a significantly higher ZO-1 concentration than that of the DSS group (*p* < 0.05) (Figure 5C,F). *D. hansenii* QSCLS6L3 did not significantly upregulate the concentrations or expression levels of these factors (*p* > 0.05). In general, *S. cerevisiae* QHNLD8L1 intervention improved the concentration and expression levels of TJPs, which may have helped to ameliorate colitis.

### 2.7. Effect of Yeast Strains on Fecal SCFAs in Colitis Mice

We measured the concentrations of fecal SCFAs (Figure 6A–F). The acetate and propionate concentrations were significantly lower in mice with colitis, but the differences in the other four SCFAs were not significant. The *S. cerevisiae* QHNLD8L1 treatment significantly upregulated the propionate and butyrate levels (*p* < 0.05). Although SCFAs in the feces of mice treated with *K. marxianus* QHBYC4L2 were increased, propionate was the only SCFAs that was increased significantly (*p* < 0.05). The valerate concentration was increased in the *S. boulardii* CNCMI-745 group significantly (*p* < 0.05), and all six SCFAs were unaffected by the *D. hansenii* QSCLS6L3 treatment significantly (*p* > 0.05), indicating that it had no preventive efficacy against colitis.

### 2.8. Effects of Yeast Strains on Gut Microbiota in Colitis Mice

According to α-diversity analysis, the gut microbiota of colitis mice showed a significantly lower observed index, which is closely related to microbial richness (*p* < 0.05) (Figure 7A). In addition, the Chao1 and Shannon index values, which reflect microbial diversity, were decreased significantly in the DSS group (*p* < 0.05). The treatments with yeasts, except for *D. hansenii* QSCLS6L3, increased the Chao1 index value (*p* < 0.05) (Figure 7B). *S. boulardii* CNCMI-745 and *S. cerevisiae* QHNLD8L1 also significantly increased the Shannon index value (*p* < 0.05) (Figure 7C). The β-diversity of gut microbiota was investigated using principal coordinate (PCoA) analysis. The mice in the DSS group had a significantly different gut microbial community from that of the control group, which was shifted to some extent by the yeast treatments, especially *K. marxianus* QHBYC4L2 and *S. cerevisiae* QHNLD8L1 (Figure 7D).

The dominant phyla were Firmicutes, Bacteroidetes, and Proteobacteria. After DSS exposure, the relative abundances of Firmicutes and Bacteroidetes decreased, whereas the relative abundance of Proteobacteria increased. However, they were differentially altered after the yeast interventions. *S. cerevisiae* QHNLD8L1 had the highest relative abundance of Firmicutes, followed by *K. marxianus* QHBYC4L2, *S. boulardii* CNCMI-745, and *D. hansenii* QSCLS6L3. Additionally, the Proteobacteria relative abundance was mostly decreased by *K. marxianus* QHBYC4L2, followed by *S. boulardii* CNCMI-745, *S. cerevisiae* QHNLD8L1, and *D. hansenii* QSCLS6L3. However, there was no obvious change in the relative abundance of Bacteroidetes among the yeast-treated groups (Figure 7E).

To further reveal the microbes among groups, we evaluated the relative abundances of microorganisms at the genus level (Figure 7F) and screened 25 differential bacteria using the linear discriminant analysis (LDA) effect size (LEfSe) (Figure 8A). Notably, compared to the control group, *Lactobacillus* was significantly reduced after DSS induction and was subsequently increased by *S. cerevisiae* QHNLD8L1 (*p* < 0.05) (Figure 8B). Cecal microbiota in colitis mice was enriched with *Escherichia*–*Shigella* and *Enterococcus*. *S. cerevisiae* QHNLD8L1 and *K. marxianus* QHBYC4L2 significantly reversed these changes, whereas *S. boulardii* CNCMI-745 downregulated the relative abundance of *Enterococcus* (*p* < 0.05) (Figure 8C,D). Moreover, *S. boulardii* CNCMI-745 and *K. marxianus* QHBYC4L2 significantly increased the relative abundances of *Parasutterella* and *Mucispirillum* in mice with colitis, respectively (*p* < 0.05) (Figure 8E,F). *Butyricimonas* and *Faecalibaculum* were significantly enriched in *S. cerevisiae* QHNLD8L1 group (*p* < 0.05) (Figure 8G,H). *Parabacteroides’* relative abundance was significantly increased by *D. hansenii* QSCLS6L3 (*p* < 0.05) (Figure 8I).

The interactions between different genera could help us to better recognize their role in DSS-induced colitis, as the gut microbiota interacts to maintain a dynamic balance. Correlation analysis between the 12 different genera showed that *Lactobacillus* was significantly negatively correlated with *Escherichia–Shigella* (*p* < 0.01) and *Parabacteroides* (*p* < 0.05). Moreover, *Butyricimonas* was significantly positive correlated with *Faecalibaculum, Mucispirillum,* and *Sphingomonas* (*p* < 0.01); *Parasutterella* was a significant negatively correlated with *Sphingomonas* (*p* < 0.01) (Figure 8J).

### 2.9. Correlation Analysis between Gut Microbiota and Colitis Indexes

A correlation analysis between 14 significantly different genera among the groups and 19 experimentally determined DSS biomarkers was conducted to demonstrate the effects of the microbiota on colitis (Figure 8K). *Escherichia–Shigella*, *Enterococcus*, and *Parabacteroides’* relative abundances were negatively correlated with body weight and colon length and positively correlated with DAI, inflammatory cytokines, and enzyme concentrations (*p* < 0.01). *Lactobacillus’* relative abundance was positively correlated with colon length and propionate and negatively correlated with DAI, inflammatory cytokines, and enzyme concentrations (*p* < 0.01). Notably, there was a significant positive correlation between the relative abundance of *Parasutterella* and *Faecalibaculum* and the levels of butyrate and IL-10 (*p* < 0.01). *Butyricimonas* and *Mucispirillum’s* relative abundances were positively correlated with acetate and butyrate (*p* < 0.05). In addition, *Butyricimonas’* relative abundance was negatively correlated with TNF-α and MPO levels (*p* < 0.05). These correlation results demonstrated DSS was involved in the development of colitis, and the yeast intervention had protective effects against colitis, except in *D. hansenii* QSCLS6L3.

### 2.10. Functional Predictions of Gut Microbiota Modified by Yeast Strains in Colitis Mice

We conducted a PICRUSt analysis with the Kyoto Encyclopedia of Genes and Genomes (KEGG) pathway (level 3) to predict functional distinction in the gut microbiota among groups. The principal component analysis (PCoA) results showed that DSS induction significantly changed the function of the gut microbiota compared with that of the control group and was subsequently altered by the yeast intervention to some extent (Figure 9A). The gene expression of 21 pathways was downregulated, whereas that of 19 pathways was upregulated after DSS exposure (*p* < 0.05) (Figure 9B). Moreover, compared to the DSS group, 35, 22, and 17 pathways were enriched in *S. boulardii* CNCMI-745, *K. marxianus* QHBYC4L2, and *S. cerevisiae* QHNLD8L1, respectively (*p* < 0.05) (Figure 9C). Additionally, the pathways involved in human papillomavirus infection and viral carcinogenesis were upregulated by *D. hansenii* QSCLS6L3 (*p* < 0.05) (Appendix A).

The pathways in the *S. cerevisiae* QHNLD8L1, *S. boulardii* CNCMI-745, *K. marxianus* QHBYC4L2, and DSS groups were selected for a further investigation. Compared to the DSS group, the genes related to pathways including fatty acid biosynthesis, porphyrin and chlorophyll metabolism, neomycin, kanamycin, gentamicin biosynthesis, and carbon metabolism were significantly upregulated by *S. cerevisiae* QHNLD8L1. Interestingly, the pathway involved in carbon metabolism was further diminished by the DSS treatment compared to that of the control group. In addition, the pathways involved in glycosaminoglycan degradation and other glycan degradation were more enriched in the *S. boulardii* CNCMI-745 and *K. marxianus* QHBYC4L2 groups compared to those in the DSS group (Appendix A).

## 3. Discussion

Accumulating evidence points to the role of gut microbes in diseases [16,17], and the effectiveness of probiotics in preventing and relieving UC recurrence has been examined [18,19,20]. Most studies on probiotic microorganisms have been conducted in bacterial systems, but in recent years, some yeasts have also shown potential [21]. Yeasts have become model organisms for many biological activities, and yeasts with derivatives or by-products have attracted a lot of interest [22]. Recently, yeasts have been increasingly thought to have benefits for disease alleviation; however, yeast strains exhibit interspecific differences in phenotypic characteristics. *S. boulardii* CNCMI-745 relieves UC through multiple mechanisms, including alterations in the intestinal microbiota, epithelial barrier defects, and immunological effects [23,24]. In addition, it is a non-pathogenic yeast strain that is safe for consumption [25]. Based on these characteristics, *S. boulardii* CNCMI-745 has often been used as a positive control strain to evaluate the potential probiotic functions of yeast species. In our study, yeast intervention, particularly with *S. cerevisiae* QHNLD8L1, effectively alleviated DSS-induced colitis, whereas *D. hansenii* QSCLS6L3 did not improve colitis in mice.

Gastrointestinal (GIT) tolerance and HT-29 cell adhesion properties are widely used in vitro to identify potential probiotic strains because they are related to strain survival and intestinal colonization [26]. Including *S. boulardii* CNCMI-745, all four yeast strains exhibited a high level of tolerance to simulated gastrointestinal juice (over 75%) and high intestinal adhesiveness (10–15%) (Table 1), which is consistent with previous research [27,28,29]. The ability of yeast strains to protect the intestinal barrier was investigated in vitro in a Caco-2 cell model stimulated with DSS. *S. boulardii* CNCMI-745 and *S. cerevisiae* QHNLD8L1 significantly increased the relative expression levels of Claudin-1 and Occludin (Figure 1). Similar to our results, *S. cerevisiae* I4 significantly upregulated these two genes in HT-29 cell monolayers [30]. Mu et al. showed that yeast metabolites promoted RAW264.7 macrophage proliferation and reduced NO production [10].

The role of different yeast strains in UC prevention remains unclear [31,32]. Hence, we evaluated the protective effects of these four yeast strains as interventional agents in mice with colitis. In agreement with previous results [33,34], mice administered 2.5% DSS showed clinical symptoms, such as weight loss, hematochezia, and colon shortening, which were effectively reversed by the yeast interventions, except for *D. hansenii* QSCLS6L3, to varying degrees (Figure 2). As demonstrated by H&E staining, epithelial structural damage, inflammatory cell infiltration, and crypt disappearance were reduced by these yeasts (Figure 3). Decreases in mucin and goblet cells are markers of IBD [35]. AB staining showed that mucin secretion and goblet cell numbers were improved, revealing that yeast therapy may delay the progression of colitis by inhibiting mucus disruption and goblet cell depletion (Figure 3). These macroscopic markers directly reflected the amelioration of colitis by the three yeast strains, *S. cerevisiae* QHNLD8L1 in particular.

Inflammation is a classic characteristic of IBD. Once the mucosal barrier is compromised, intestinal epithelial cells rapidly release pro-inflammatory cytokines, which cause enormous inflammation and tissue injury [36]. TNF-α can result in mucosal inflammation and activate the NF-κB pathway; meanwhile, NF-κB may stimulate the release of TNF-α, IL-1β, and IL-6 [37]. IL-10 is crucial for host infection because it limits the immune response to pathogens and inhibits pro-inflammatory cytokine production [38,39]. In our study, after DSS exposure, pro-inflammatory cytokine concentrations were higher and anti-inflammatory cytokine concentrations were lower. Yeast treatments, except for *D. hansenii* QSCLS6L3, could restore these changes in colitis (Figure 4). In particular, *S. cerevisiae* QHNLD8L1 could significantly decrease TNF-α and increase IL-10 concentrations, which was consistent with *S. cerevisiae* JKSP39 in previous reports [15]. MPO activity is a classic biomarker used to evaluate inflammation in colitis and reflects the degree of neutrophil infiltration [40]. INOS is produced in response to activation and stimulation by microorganisms, cytokines, and macrophages [41]. The overexpression of COX-2 in inflammatory cells may promote colitis progression [42]. The levels of these inflammatory enzymes were raised after DSS exposure. *S. boulardii* CNCMI-745 and *S. cerevisiae* QHNLD8L1 administration significantly inhibited the increase in MPO and iNOS activities (Figure 4), indicating that they promoted inflammatory remission in mice with DSS-induced colitis.

Inflammation may damage the intestinal barrier, and barrier dysfunction can result in severe bacterial inflammation. An appropriate intestinal barrier can limit antigen invasion and prevent abnormal immune reactions [43]. Thus, maintaining a tight and intact gut barrier is a major goal of IBD therapy. TJPs, such as transmembrane Claudins, Occludin, and intracellular ZO-1, are among the most important junction structures connecting intestinal epithelial cells [44]. In a previous study, some yeasts were reported to reduce gut barrier disruption by increasing the relative expression of Occludin and ZO-1, which relieved DSS-induced colitis [45]. Our results showed that the expression levels and concentrations of Claudin-1 and ZO-1 were increased by *S. cerevisiae* QHNLD8L1, which were similar to those of the control group and even higher than those subjected to the *S. boulardii* CNCMI-745 treatment (Figure 5), suggesting that *S. cerevisiae* QHNLD8L1 may attenuate colitis by improving TJPs.

The gut microbiota protects the intestinal epithelium and mucosa and keeps pathogens out of the gastrointestinal tract. Thus, the pathogenesis of IBD may be largely influenced by gut microbes. The disruption of gut microbial composition may affect the mucosal environment and intestinal barrier, which in turn affects gut microbes themselves [46]. Our study demonstrated that DSS reduces gut flora diversity and alters gut flora composition, resulting in intestinal microecological disruption in mice. After DSS exposure, the microbiota species richness was significantly increased by yeasts other than *D. hansenii* QSCLS6L3. Moreover, *K. marxianus* QHBYC4L2 and *S. cerevisiae* QHNLD8L1 caused a certain degree of change in the gut microbiota toward that of the control group (Figure 7). Proteobacteria, a microbial indicator of gut flora imbalance [47], was enriched in the DSS group. After yeast intervention, the relative abundance of Proteobacteria decreased, and that of Firmicutes increased to some degree (Figure 7). Therefore, these yeasts reduced the microbiota structure imbalance, which is consistent with the results of a previous study [30].

The microbial community of DSS-induced mice was disrupted, as revealed by the increase in the relative abundances of *Escherichia–Shigella* and *Enterococcus* (Figure 8). As typical pathogenic bacteria, *Escherchia–Shigella* is commonly seen in patients with colitis and colorectal cancer; increased *Enterococcus* could exacerbate intestinal inflammation as well [48]. We discovered that *Escherichia–Shigella* and *Enterococcus* were negatively correlated with Occludin and positively correlated with pro-inflammatory factors. The predicted gut microbiota function results showed that DSS could alter gut microbiota function, specifically by lowering the metabolic rate in colitis mice (Figure 9). *S. cerevisiae* QHNLD8L1 restored the altered levels of these two genera and promoted the growth of *Lactobacillus* (Figure 8), which has been reported to reduce DSS-induced inflammation and damage to the colons of mice [49]. Our correlation results showed that *Lactobacillus* was negatively corrected with *Escherchia–Shigella* and pro-inflammatory factors and positively corrected with colon length in mice.

In addition, the *S. cerevisiae* QHNLD8L1 treatment significantly enhanced the growth of protective strains such as *Butyricimonas* (Figure 8), and the result was similar to the improvement of galangin against colitis by promoting *Butyricimonas* [50]. A previous study indicated that *Butyricimonas’* relative abundance was lower in patients with CD than it was in healthy individuals [51]. Our analysis showed that *Butyricimonas* had a negative correlation with proinflammatory cytokines, such as TNF-α and MPO, indicating that it may be effective in the progression of colitis. *S. cerevisiae* QHNLD8L1 significantly increased the relative abundance of *Faecalibaculum* (Figure 8), a novel generation of potentially beneficial microorganisms that can produce SCFAs. According to a recent report, *Faecalibaculum* reduces inflammation by stimulating the development of Tregs in the colon, and a decrease in *Faecalibaculum* levels is associated with the emergence of colorectal cancer [52]. Hu et al. [53] demonstrated that VK2 ameliorates DSS-induced colitis and reduces intestinal microflora dysbiosis by increasing *Faecalibaculum* and promoting SCFA production. The increased number of *Faecalibaculum* genera motivated a rise in the level of *Butyricimonas*, corroborating our finding that *Faecalibaculum* had a positive correlation with *Butyricimonas*.

As the main fermentation products of non-digestible dietary polysaccharides produced by colonic microorganisms, SCFAs are considered to be regulators of intestinal function and inflammatory responses [54]. In this study, most SCFAs decreased after DSS exposure, and *S. cerevisiae* QHNLD8L1 increased propionate and butyrate levels, which agreed with the changes in the gut microbiota (Figure 6). Propionate activates GPR43 to promote Foxp3 expression and induces Treg proliferation, thereby enhancing IL-10 secretion [55]. Butyrate prevents mucosal inflammation by increasing the IL-10 levels, and its nutritional effects on epithelial cells are well understood [56]. Further analysis of the alterations in gut microbiota function revealed that fatty acid biosynthesis and carbon metabolism were enhanced after the *S. cerevisiae* QHNLD8L1 treatment (Figure 9). The levels of SCFAs, including acetate, propionate, and butyrate, were higher in mice treated with yeast-derived complex dietary polysaccharides, which promote gut immune regulation [57]. Polysaccharides and polypeptides of *S. boulardii* have also been reported to alleviate colitis by accelerating the growth of certain probiotics and generating microbial metabolite SCFAs [58]. In summary, *S. cerevisiae* QHNLD8L1 suppresses colitis development by limiting pathogen colonization and increasing the number of beneficial microorganisms that help to restore dysfunction.

In addition, the *S. boulardii* CNCMI-745 treatment significantly increased the relative abundance of *Parasutterella* (Figure 8), which contributes to bile acid metabolism. *Parasutterella* has a positive correlation with butyrate and IL-10, indicating that it can relieve colitis by altering the metabolic products of microbes. *Mucispirillum,* similar to *Akkermansia*, colonizes the mucus layers associated with gut inflammation and has a special system to manage oxidative stress in colitis [35,59]. *K. marxianus* QHBYC4L2 significantly upregulates *Mucispirillum* abundance (Figure 8), which is consistent with diosgenin-attenuated colitis caused by increasing *Mucispirillum* levels [60]. However, it has also been found that fructooligosaccharide alleviates gut inflammation and decreases *Mucispirillum* [39]. In addition, in terms of the functional prediction of intestinal flora, the pathways related to glycosaminoglycan degradation and other glycan degradations were enriched by *K. marxianus* QHBYC4L2 (Figure 9). Glycosaminoglycans are degraded by gut-colonizing microbes and the levels of it are reduced in individuals with IBD [61]. Hence, the role of *Mucispirillum* in colitis requires further investigation. *D. hansenii* QSCLS6L3 significantly upregulated *Parabacteroides’* relative abundance, which was more abundant in IBD patients and in a mouse model [62] and worsened the colitis [63], confirming our findings. The differences in the dominant intestinal flora alterations and functional pathway modulation may explain the alleviative effects of selected yeast strains, especially *S. cerevisiae* QHNLD8L1, on colitis.

## 4. Materials and Methods

### 4.1. Yeast Strains Isolation, Purification and Identification

*K. marxianus* QHBYC4L2, *S. cerevisiae* QHNLD8L1, and *D. hansenii* QSCLS6L3 were isolated from fermented Chinese foods using a standard methodology described by Tikka et al. [64]. Serial dilutions were prepared and inoculated on yeast extract peptone dextrose (YPD) agar plates at 28 °C for 48 h. After incubation, colonies were selected based on morphology and sub-cultured on YPD agar plates for further purification. The commercial probiotic yeast *S. boulardii* CNCMI-745 isolated from BIOFLOR (BIOCODEX, Gentilly, France) under the same conditions was used as a positive control. Yeast strains were identified by amplification of the D1/D2 domain of 26S rDNA region using primers NL-1 (5′-GCATATCAATAAGCGGAGGAAAAG-3′) and NL-4 (5′-GGTCCGTGTTTCAAGACGG-3′). Subsequently, amplicons were sent to GENEWIZ (Suzhou, China) for sequencing, and species were identified using National Library for Biotechnology Information (NCBI). All strains were prepared in YPD liquid medium at 10^8^ CFU/mL. After three generations of activation, the medium was centrifuged (4 °C, 5000× *g*, 10 min) to obtain the sediment and frozen in 30% glycerol solution at −80 °C for following experiments.

### 4.2. Gastrointestinal Transit Tolerance Assay

To determine the GIT survival rate, each yeast strain was treated with simulated gastric fluid (3 g/L pepsin (1:10,000), pH 3.0, 3 h) and intestinal juice (1 g/L trypsin (1:250) and 3 g/L bile salt, pH 8.0, 4 h)) as previously described [9]. After centrifugation at 5000× *g* for 10 min (4 °C), the initial concentration of the yeasts reached 10^8^ CFU/mL at 0 h. Strain viable numbers were tested by plate counting at 3 and 7 h.

### 4.3. Adhesion Ability Assay

The yeast adhesion capacity was evaluated using a modified method similar to that used by Meng et al. [30]. Human colon cell line HT-29 was purchased from American Type Culture Collection (Manassas, VA, USA). HT-29 cells were inoculated with 10^5^ cells/well in 12 well plates for 48 h. Yeast strains were suspended in a blank DMEM medium and prepared at 1 × 10^7^ CFU/mL. After co-culturing with yeast for 4 h, HT-29 cells were washed to remove any unbound yeast. Finally, adherent yeasts were detected by plate counting to determine the CFU number after dilution.

### 4.4. DSS-Induced Injury in Caco-2 Cells

The relative expression of tight junction proteins was determined as described by Yu et al. [26]. Human colon cell line Caco-2 was purchased from American Type Culture Collection (Manassas, VA, USA). Caco-2 cells were inoculated with 2 × 10^5^ cells/well in 6 well plates for 48 h, and the cells were treated with DSS (2.5%) to initiate intestinal barrier damage. Meanwhile, 100 μL yeast strains (10^7^ CFU/mL) were co-cultured with cells for 12 h. Treated Caco-2 cells were washed twice and separated from the culture plates.

Total RNA extraction and cDNA reverse transcription were performed using a FastPure Cell Total RNA Isolation Kit and an HiScript III RT SuperMix for qPCR, respectively (Vazyme Co., Ltd., Nanjing, China). RT-qPCR was conducted using a Fast SYBR Green Master Mix with a StepOne™ real-time PCR system with a BioRad-CFX384 machine (Bio-Rad Co., Ltd., Hercules, CA, USA). The primer sequences are listed in Appendix A. The relative expression level of relative genes was calculated by 2^−ΔΔCT^.

### 4.5. Animal Experiment Design on Colitis

Six-week-old male SPF C57BL/6N mice (18–22 g) were supplied by Vital River Co., Ltd. (Shanghai, China). All experimental procedures were approved by the Experimental Animal Ethics Committee of Jiangnan University (approval number: JN. No 20211215c1100415[534]). Mice were housed under standard conditions (22~24 °C, 40~70% relative humidity, 12 h photoperiod) and given water and chow for one week before the experiments began.

Forty-eight mice were randomly divided into six groups (n = 8) after one week of adaptation. According to Appendix A, the experimental period was two weeks in total; mice in the control and model groups orally received 200 µL of saline, while mice in the four yeast groups were gavaged with 200 µL of the corresponding yeast suspension, respectively. In the second week, mice in all groups were administered 2.5% (*w*/*v*) DSS in water, except for the control group.

### 4.6. Assessment of Colitis Symptoms

Body weight, stool consistency, occult blood, and DAI were recorded daily based on the scoring principles shown in Appendix A during DSS exposure [65]. The colon length of the mice was determined after euthanasia. Colon tissues (0.5 cm) were fixed in 4% (*w*/*v*) formalin solution, dehydrated in ethanol, embedded in paraffin, sliced, and stained with H&E. Colonic histological severity was scored from four perspectives: inflammatory condition, mucosal injury, crypt damage, and lesion degree [66]. The degree of colonic mucous layer injury was determined by a modified method of Steedman [67], and mucous epithelial thickness and goblet cell numbers were measured using AB staining.

### 4.7. Biochemical Assays

Colon tissues were homogenized with saline and centrifuged to collect the supernatant. According to the manufacturer’s instructions, TNF-α, IL-6, IL-10, and IL-1β were measured using enzyme-linked immunosorbent assay (ELISA) kits (R&D Systems Co., Ltd., Shanghai, China). The concentrations of MPO, COX-2, iNOS, Claudin-1, Occludin, and ZO-1 were determined using mouse-specific ELISA kits (SenBeiJia Co., Ltd., Nanjing, China). Total protein content was determined using a bicinchoninic acid (BCA) protein assay kit (Beyotime Co., Ltd., Shanghai, China).

### 4.8. Tight Junction Protein Expression in Colon

Total RNA in the colon was extracted using Trizol reagent, and the following experimental procedures were performed as described in Section 4.4. Sequences of the primer are shown in Appendix A.

### 4.9. Short-Chain Fatty Acid Measurement in Feces

The concentrations of SCFAs (acetate, propionate, butyrate, isobutyrate, valerate, and isovalerate) in feces were determined according to the previous method [40]. Samples (20 mg) were resuspended into 500 μL saturated NaCl solution, and then 20 μL sulfuric acid (10%) was added for acidification. SCFAs were extracted by adding 1000 μL diethyl ether. The mixture was centrifuged (12,000× *g*, 4 °C, 15 min), and the aqueous content was removed by adding 0.25 g Na2SO4. SCFAs were quantified by gas chromatography–mass spectrometry (GC-MS) (Shimadzu Corporation, Tokyo, Japan) using the external standard method.

### 4.10. Gut Microbial and Bioinformatics Analysis

Sequencing of the gut microbiota composition was performed according to Chen et al. [68]. DNA was extracted using the FastDNA Spin Kit (MP Biomedicals, Carlsbad, CA, USA). The V3-V4 region of the 16S rRNA was amplified using primers 341F and 806R. PCR product purification, quantification, library generation, and sequencing on Illumina MiSeq were performed as previously described by Yan et al. [69].

Microbiota analysis of the raw sequencing data was performed using the QIIME2 pipeline. Briefly, the α-diversity, including the Observed, Chao1, and Shannon indices, was evaluated by species diversity and distributional uniformity among OTUs. PCoA analysis displayed β-diversity by Bray–Curtis distances. Microbiological markers were separated using the LEfSe method. The association network was analyzed using Spearman’s correlation coefficients. Functional pathway prediction of the microbiota was performed using PICRUSt analysis.

### 4.11. Statistical Analysis

SPSS26.0 and GraphPad Prism 9.0 were conducted for data analysis and plotting. All the ultimate results are represented as the mean ± SEM for each group. Significant differences were evaluated using ANOVA and Tukey’s multiple comparison test, with a significance threshold of *p* < 0.05.

## 5. Conclusions

In summary, our study demonstrated that the selected yeasts possessed a good GIT tolerance, adhesion ability, and intestinal barrier enhancement in vitro. Especially, *S. cerevisiae* QHNLD8L1 alleviated the DSS-induced colitis symptoms by a mechanism primarily including modulating the inflammatory factors, enhancing the gut barrier, regulating specific microbiota and strengthening their metabolic functions, and increasing SCFA production. *S. cerevisiae* QHNLD8L1 supplements could be a possible pathway for UC therapy, which offers a new direction to obtain more beneficial yeasts for human health. However, the effectiveness and safety of the yeast in human settings are required to conduct future experiments.

## Figures and Tables

**Figure 1 ijms-24-06721-f001:**
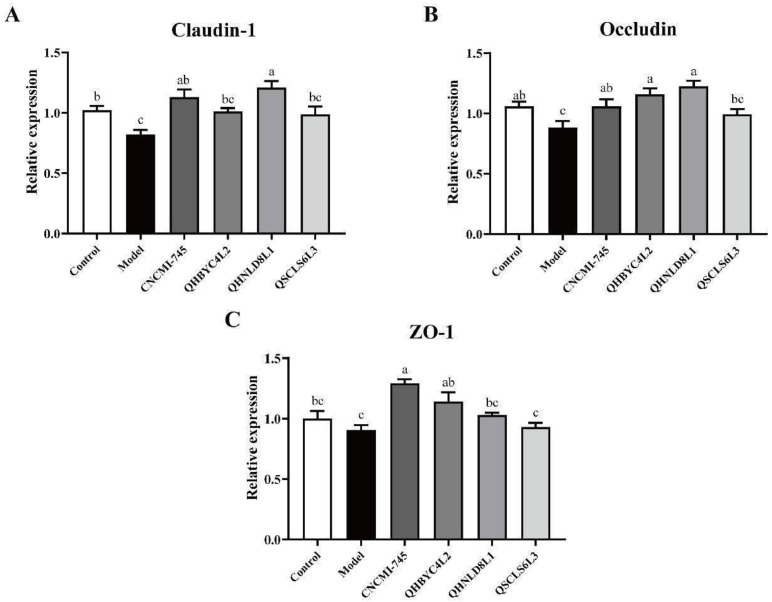
Effect of yeasts on the relative expression of tight junction proteins in Caco-2 cells. (**A**) Claudin-1. (**B**) Occludin. (**C**) ZO-1. The results are presented as mean ± SEM. Values with different letters a–c represent significant differences at *p* < 0.05.

**Figure 2 ijms-24-06721-f002:**
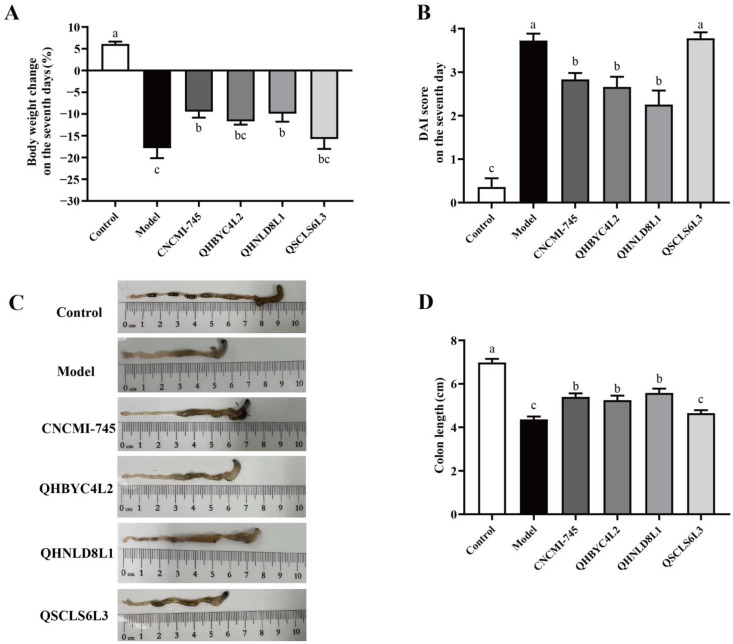
Effect of yeast strains on colitis symptoms. (**A**) Body weight change (%). (**B**) Disease activity index (DAI) score. (**C**) Macroscopic photos of colonic tissues. (**D**) Colon length (cm). The results are presented as mean ± SEM. Values with different letters a–c represent significant differences at *p* < 0.05.

**Figure 3 ijms-24-06721-f003:**
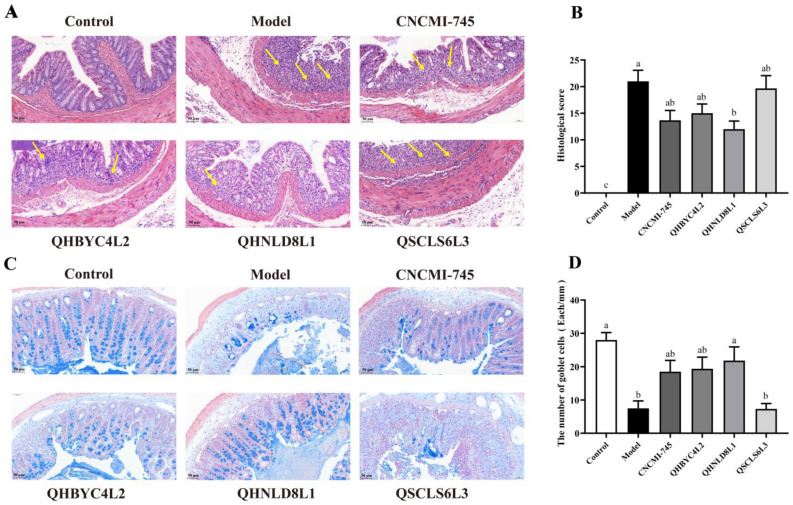
Effect of yeasts on colon histological structure and intestinal mucosa. (**A**) Hematoxylin and Eosin (H&E) stained histological sections (magnification, ×20). (**B**) Histopathology scores. (**C**) Alcian Blue (AB) stained pathological section (magnification, ×20). (**D**) Goblet cell number. The results are presented as mean ± SEM. Values with different letters a–c represent significant differences at *p* < 0.05. The yellow arrows represent crypt structure destruction and inflammatory cell infiltration.

**Figure 4 ijms-24-06721-f004:**
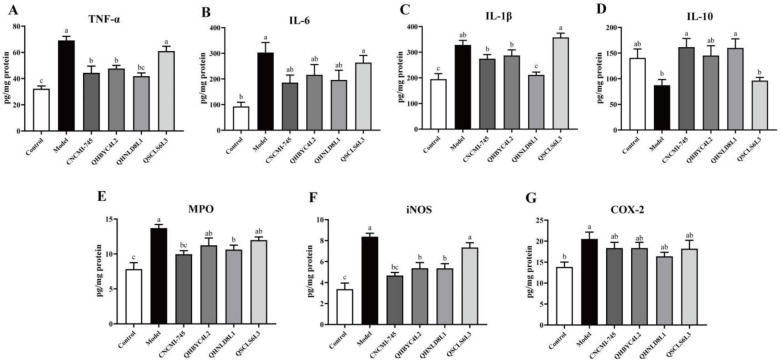
Effect of yeasts on inflammatory cytokine and enzyme concentrations. (**A**) TNF-α. (**B**) IL-6. (**C**) IL-1β. (**D**) IL-10. (**E**) MPO. (**F**) iNOS. (**G**) COX-2. The results are presented as mean ± SEM. Values with different letters a–c represent significant differences at *p* < 0.05.

**Figure 5 ijms-24-06721-f005:**
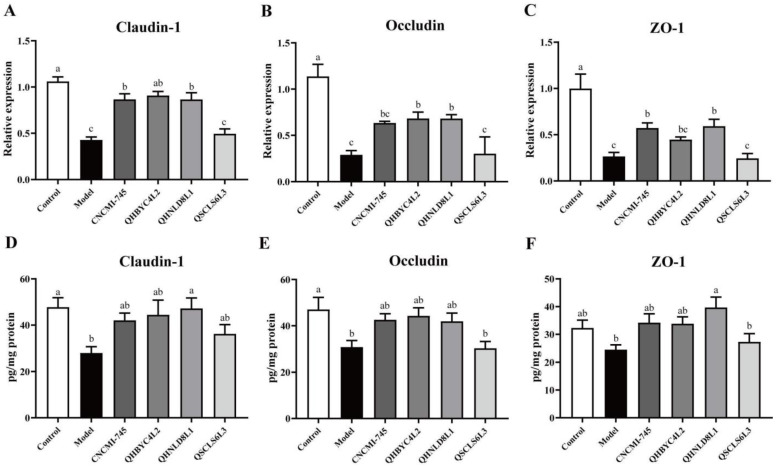
Effect of yeasts on intestinal barrier protection. (**A**–**C**) Relative expression levels of Claudin-1, Occludin, and ZO-1. (**D**–**F**) Concentrations of Claudin-1, Occludin, and ZO-1. The results are presented as mean ± SEM. Values with different letters a–c represent significant differences at *p* < 0.05.

**Figure 6 ijms-24-06721-f006:**
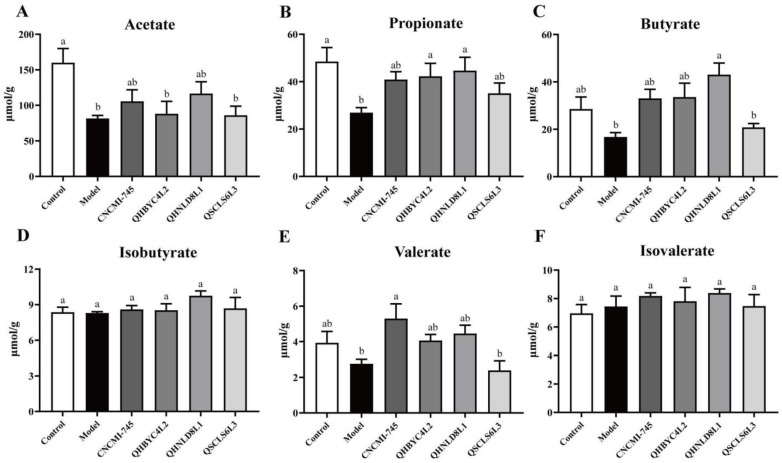
Effect of yeasts on short chain fatty acids (SCFAs) concentrations. (**A**) Acetate. (**B**) Propionate. (**C**) Butyrate. (**D**) Isobutyrate. (**E**) Valerate. (**F**) Isovalerate. The results are presented as mean ± SEM. Values with different letters a–b represent significant differences at *p* < 0.05.

**Figure 7 ijms-24-06721-f007:**
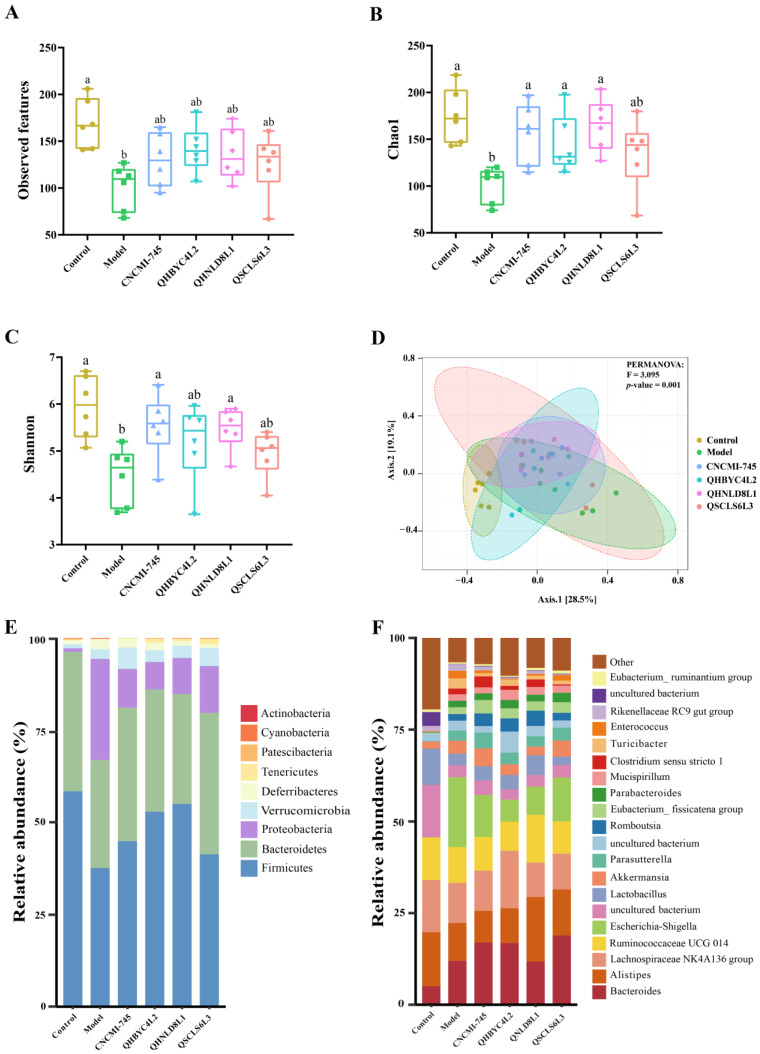
Effect of yeasts on gut microbiota diversity and phylum/genus level changes. (**A**–**C**) α-diversity analysis: Observed, Chao1 index, and Shannon indices. (**D**) Principal coordinates analysis (PCoA) of β-diversity. (**E**) A pile-up histogram at the phylum level. (**F**) A pile-up histogram at the genus level. The results are presented as mean ± SEM. Values with different letters a–b represent significant differences at *p* < 0.05.

**Figure 8 ijms-24-06721-f008:**
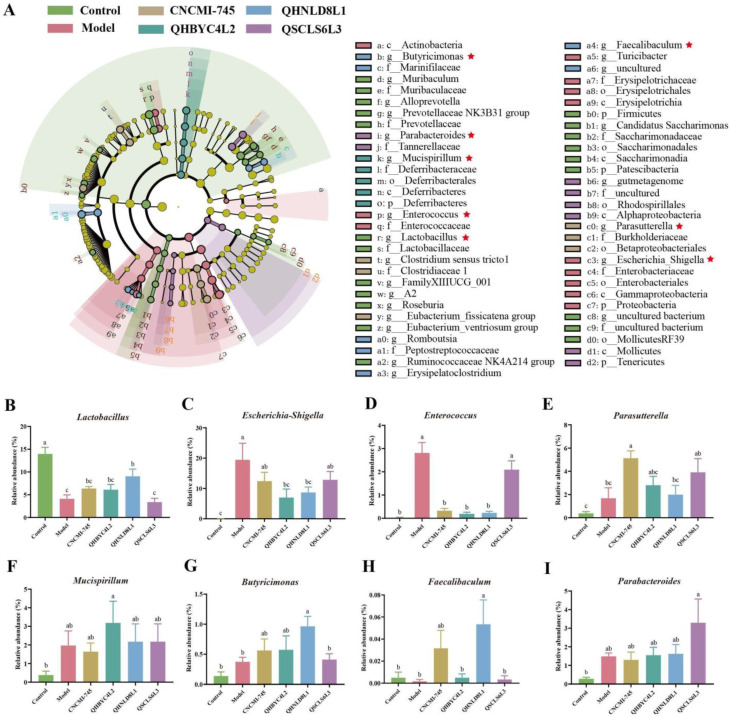
Effect of yeasts on dominant microorganisms. (**A**) Linear discriminant analysis (LDA) effect size (LEfSe) of the genus with a significant difference among groups at *p* < 0.05 and log LDA > 3.0. Genera with significant differences among groups are marked with the red stars. (**B**–**I**) Eight significant genera: *Lactobacillus*, *Escherichia–Shigella*, *Enterococcus*, *Parasutterella*, *Mucispirillum*, *Butyricimonas*, *Faecalibaculum,* and *Parabacteroides*. (**J**) Correlation analysis of 12 significantly different genes. (**K**) Correlation analysis between 19 DSS biomarkers and the 12 different genes. The results are presented as mean ± SEM. Values with different letters a–c represent significant differences at *p* < 0.05. *: *p* < 0.05, **: *p* < 0.01, ***: *p* < 0.001, ****: *p* < 0.0001.

**Figure 9 ijms-24-06721-f009:**
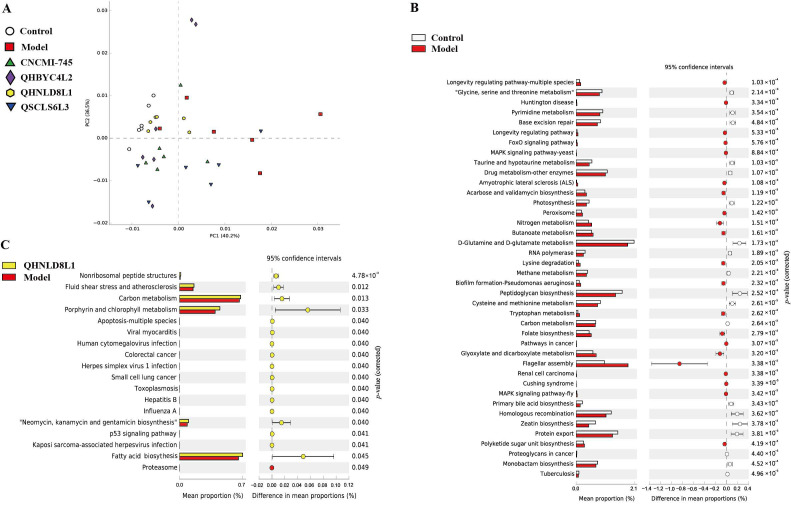
Effects of yeast strains on gut microbial functions. (**A**) Principal component analysis (PCoA) of gut microbiota function prediction in the Kyoto Encyclopedia of Genes and Genomes (KEGG) pathways at level 3. (**B**) Differences in predicted functions between the control and DSS groups. (**C**) Differences in predicted functions between *S. cerevisiae* QSCLS6L3 and DSS groups.

**Table 1 ijms-24-06721-t001:** Survival and adhesion rates of yeast strains.

Strains	Survival Rate (%)		Adhesion Rate (%)
	Gastric Juice	Intestinal Juice	
*S. boulardii* CNCMI-745	89.08 ± 0.513 ^a^	78.30 ± 0.444 ^b^	13.79 ± 0.158 ^a^
*K. marxianus* QHBYC4L2	86.52 ± 0.538 ^b^	79.86 ± 0.516 ^b^	12.69 ± 0.391 ^a^
*S. cerevisiae* QHNLD8L1	87.15 ± 0.462 ^b^	82.93 ± 0.262 ^a^	13.93 ± 0.511 ^a^
*D. hansenii* QSCLS6L3	84.29 ± 0.409 ^c^	76.92 ± 0.538 ^c^	10.46 ± 0.627 ^b^

Letters a–c represent significant differences at *p* < 0.05.

## Data Availability

The datasets generated during and/or analyzed during the current study are either shown in the manuscript and Appendix A or available from the corresponding author upon reasonable request.

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
