# Peer review of "Anti-Inflammatory, Barrier Maintenance, and Gut Microbiome Modulation Effects of Saccharomyces cerevisiae QHNLD8L1 on DSS-Induced Ulcerative Colitis in Mice"

_ijms, 2023, doi:10.3390/ijms24076721_

Round 1

Reviewer 1 Report

This manuscript describes studies designed to test the hypothesis that species of yeast can act as probiotics to ameliorate colitis in a mouse model.  The authors show that several yeast strains, and especially Saccharomyces cerevisiae QHNLD8L1, improved several parameters of colitis.  The effect appears to be through modulation of DSS colitis-induced changes in the bacterial microbiome.  The manuscript is well written and the data are presented clearly.  Attention the following comments would strengthen the impact of the work.

The authors have entitled section 2.6, “Effect of yeast strains on intestinal barrier function in colitic mice”.  In fact, the authors have not measured barrier function and are making an inference based on changes in the expression of tight junction proteins.  Increased tight junction protein expression usually, but not always, correlates with improved barrier function, but only if the increased protein is localized to the tight junction and does not remain in the cytoplasm.  Ideally the authors would conduct studies to measure barrier function (ie lumen-to-blood flux of a marker such as FITC-dextran) or at least change the wording in this section.   Furthermore, in the Methods section 4.4, the authors refer to studies in DSS-treated Caco-2 cells, but the results of these studies do not appear in the results section.  Clarification is required. 

Overall, the methods section lacks detail and should be expanded.  One example is in section 4.1 (yeast strains and preparation).  Exactly how were the yeast strains isolated from the food?  How were they purified and identified?  The authors do not mention the source of Saccharomyces bourdii.  Other sections of the methods could similarly be expanded to provide the reader with more details of the procedures that were used. 

The authors use the lettering method to indicate significant differences among groups.  Some readers may not have seen this method before.  The authors should explain in the figure legends that bars with the same letter are not significantly different from each other. 

Author Response

Response to Reviewer 1 Comments:

Reviewer #1:

We would like to appreciate you for spending your valuable time reviewing this manuscript and presenting good comments for improving the content. We are very grateful for your rigorous considerations and professional proposals. We made an effort to modify the manuscript based on your noteworthy comments. We have marked the changed place as RED and modified it in the manuscript using the “Track Changes” function. We would mark the line number of the revised manuscript in each answer.

Point 1: The authors have entitled section 2.6, “Effect of yeast strains on intestinal barrier function in colitic mice”. In fact, the authors have not measured barrier function and are making an inference based on changes in the expression of tight junction proteins. Increased tight junction protein expression usually, but not always, correlates with improved barrier function, but only if the increased protein is localized to the tight junction and does not remain in the cytoplasm. Ideally the authors would conduct studies to measure barrier function (ie lumen-to-blood flux of a marker such as FITC-dextran) or at least change the wording in this section.

Response 1: We appreciate the reviewer’s helpful comments here. We apologize for our careless mistake and realized that concluding altered intestinal barrier function was not rigorous based on changes in tight junction proteins. The wording in results section 2.6 was revised at line 160, and the wording in methods section 4.8 was also revised at line 509.

Point 2: In the Methods section 4.4, the authors refer to studies in DSS-treated Caco-2 cells, but the results of these studies do not appear in the results section. Clarification is required.

Response 2: We appreciate the reviewer’s helpful comments here.. The results corresponding to Methods section 4.4 in this article should be in section 2.2.. To make our results more straightforward, we have also added a connective sentence: “To assay the modulation of DSS-induced responce by different yeasts, we added yeast strains (107 CFU/mL) to the Caco-2 cells model co-cultured with 2.5% DSS for 12 h.” Please see line 77-87. We have double-checked the full text to make sure all results corresponded to the methods in our study.

Point 3: the methods section lacks detail and should be expanded. One example is in section 4.1 (yeast strains and preparation). Exactly how were the yeast strains isolated from the food? How were they purified and identified? The authors do not mention the source of Saccharomyces bourdii. Other sections of the methods could similarly be expanded to provide the reader with more details of the procedures that were used.

Response 3: We appreciate the reviewer’s helpful comments here. We apologize for not making the methods section detailed. We have expanded some language to better describe the main points of the experimental procedures in section 4 Materials and Methods section. We have also added two references [64] and [67]. Take section 4.1 for example, the details related to isolation, purification and identifion of yeast strains, and the information about source of Saccharomyces bourdii CNCMI-745 have been added to section 4.1. Please see line 434-446 and line 440-442.

Point 4: The authors use the lettering method to indicate significant differences among groups. Some readers may not have seen this method before. The authors should explain in the figure legends that bars with the same letter are not significantly different from each other.

Response 4: We appreciate the reviewer’s helpful comments here. We apologize for not making the significant difference representation among groups clear. We have revised our figure legends section related to the lettering method to indicate significant differences according to your suggestions.

Reviewer 2 Report

REVIEW

Dear authors,

Please consider the following comments to improve the content of your manuscript before publication. 

The work is of great application importance, because it proposes an alternative probiotic treatment different from those traditionally used with bacteria and the risks involved with the horizontal transfer of antibiotic resistance genes. Yeasts have been rarely used for these purposes, as a result of the limited research with these microorganisms, however, studies as complete as the one developed in this work put into context evidence of beneficial effects on a disease that affects millions of people in the world. such as ulcerative colitis.

The focus on changes in the intestinal microbiota and the correlation with metabolism, support the multiple mechanisms by which yeasts can provide benefits in this condition.

Without a doubt, it is a work that enriches and contributes to research on probiotics and its potential application. I consider that the article has what it takes to be published in the format presented, however, it is necessary to make some corrections along the lines indicated:

Line 4: specify in the title, the model in which the colitis developed (in vitro, in vivo, clinical).

Figure 3: aumentar el tamaño de las micrografías de los cortes histológicos (Figure 3a y 3c), así como cambiar el color de las flechas para hacer contraste ya que no se logran identificar.

Line 130: correct the word “Hstopatology”.

Line 131: correct the word “Gblet”.

Line 131: homogenize the abbreviation “AB-PAS”.

Lines 190, 192: capitalize “Chao1”.

Line 192: separate the words “D.hansenii”.

Lines 259, 278: homogenize the abbreviation “principal component analysis (PCoA)”.

Line 372: the word “function” has a different size than text.

Line 419: the word “FOS” not mentioned previously, please write the full meaning in the text.

Line 439, 463: the degree symbol is in a smaller size “°C”.

Line 517: italicize the word “in vitro”.

Lines 525, 527: homogenize to “RT-qPCR”.

Line 526: correct the word“diease”.

Line 527: correct the word “colitic”.

REFERENCES

Lines 552, 554, 556, 562, 574, 577, 589, 593, 596, 597, 607, 613, 625, 640, 643, 645, 647, 648, 652, 654, 655, 665, 688, 695, 698: abbreviate the name of the journal.

Lines 567, 569, 573, 576, 578, 580, 581, 584, 590, 592, 594, 595, 601, 605, 608, 611, 612, 613, 616, 624, 625, 628, 634, 650, 651, 657, 658, 660, 668, 672, 681, 684, 685, 691, 692, 694, 696, 697, 699, 700, 702, 706: write in italics the name of the microorganisms.

Line: 581: italicize the term “in vivo”.

Line 612: italicize the term “in vitro”.

Check the references since the number of the journal should be removed, the volume of the journal should be written in italics and in some references the number of pages is missing.

SUPPLEMENTARY INFORMATION

Lines 3, 5, 9, 20: homogenize to “RT-qPCR”.

Lines 4, 13: correct the word “diease”.

Lines 5, 20: correct the word “colitic”.

Table S1: correct gene name “Zo-1”.

Table S2: lowercase the name of the species “K. marxianus, D. hansenii”.

Please amend the requested comments and submit the revision file.

Author Response

Response to Reviewer 2 Comments:

Reviewer #2:

We would like to appreciate you for spending your valuable time reviewing this manuscript and presenting good comments for improving the content. We are very grateful for your rigorous considerations and professional proposals. We made an effort to modify the manuscript based on your noteworthy comments. We have marked the changed place as RED and modified it in the manuscript using the “Track Changes” function. We would mark the line number of the revised manuscript in each answer.

Point 1: specify in the title, the model in which the colitis developed (in vitro, in vivo, clinical).

Response 1: We appreciate the reviewer’s helpful comments here. As you believe, we forgot to add the information about model in which the colitis developed. We established a colitis model in vivo by mice induced by DSS. Thank you for reminding us. The relevant information about the colitis model used in our study was revised at line 4.

Point 2: aumentar el tamaño de las micrografías de los cortes histológicos (Figure 3a y 3c), así como cambiar el color de las flechas para hacer contraste ya que no se logran identificar.

Response 2: We appreciate the reviewer’s helpful comments here. We apologize that the scale of the micrographs of histological sections (Figure 3a and 3c) was not right due to our careless mistake. We have corrected scale “400 μm” to “50 μm”. We have also changed the color of arrow from red to yellow (figure 3a) to be better recognized according to your suggestions.

Point 3: correct the word “Hstopatology” and “Gblet”.

Response 3: We appreciate the reviewer’s helpful comments here. We apologize for our careless mistake and the word spelling was revised at line 134 and line 135.

Point 4: homogenize the abbreviation “AB-PAS”.

Response 4: We appreciate the reviewer’s helpful comments here. We agree with your point that “AB/PAS” is incorrect here and should be corrected to “AB-PAS”. Actually, we discovered the description of staining method in this experiment was wrong due to our carelessness mistake. We have corrected staining method “AB/PAS” to “AB”. Please see line 123, 135, 322 and 500. We have double-checked the full text to make sure that the experimental method used in our study was right.

Point 5: capitalize “Chao1”.

Response 5: We appreciate the reviewer’s helpful comments here. We apologize for our careless mistake and the word format was revised at line 195 and line 197.

Point 6: separate the words “D.hansenii”.

Response 6: We appreciate the reviewer’s helpful comments here. We apologize for our careless mistake and the word format was revised at line 197.

Point 7: homogenize the abbreviation “principal component analysis (PCoA)”.

Response 7: We appreciate the reviewer’s helpful comments here. We apologize for our careless mistake and the abbreviation spelling was revised at line 265 and line 284.

Point 8: the word “function” has a different size than text.

Response 8: We appreciate the reviewer’s helpful comments here. We apologize for our careless mistake and the word size was revised at line 378.

Point 9: the word “FOS” not mentioned previously, please write the full meaning in the text.

Response 9: We appreciate the reviewer’s helpful comments here. We apologize for our careless mistake and the word spelling was revised at line 424.

Point 10: the degree symbol is in a smaller size “°C”.

Response 10: We appreciate the reviewer’s helpful comments here. We apologize for our careless mistake and the degree symbol format was revised at line 455 and line 483.

Point 11: italicize the word “in vitro”.

Response 11: We appreciate the reviewer’s helpful comments here. We apologize for our careless mistake and the word format was revised at line 542.

Point 12: homogenize to “RT-qPCR”.

Response 12: We appreciate the reviewer’s helpful comments here. We apologize for our careless mistake and the word spelling was revised at line 550 and line 552.

Point 13: correct the word “diease” and “colitic”。

Response 13: We appreciate the reviewer’s helpful comments here. We apologize for our careless mistake and the word spelling was revised at line 551 and line 552.

Point 14: REFERENCES:lines 552, 554, 556, 562, 574, 577, 589, 593, 596, 597, 607, 613, 625, 640, 643, 645, 647, 648, 652, 654, 655, 665, 688, 695, 698: abbreviate the name of the journalï¼›lines 567, 569, 573, 576, 578, 580, 581, 584, 590, 592, 594, 595, 601, 605, 608, 611, 612, 613, 616, 624, 625, 628, 634, 650, 651, 657, 658, 660, 668, 672, 681, 684, 685, 691, 692, 694, 696, 697, 699, 700, 702, 706: write in italics the name of the microorganismsï¼›italicize the term“in vivo”and“in vitro”; check the references since the number of the journal should be removed, the volume of the journal should be written in italics and in some references the number of pages is missing.

Response 14: We appreciate the reviewer’s helpful comments here. We apologize for our careless mistake, and we carefully checked the reference format and fixed the format of the references cited in our paper accordingly based on your suggestions.

Point 15: SUPPLEMENTARY INFORMATION:homogenize to “RT-qPCR”ï¼›correct the word “diease”ï¼›correct the word “colitic”ï¼›correct gene name “Zo-1”and lowercase the name of the species K. marxianus, D. hansenii”.

Response 15: We appreciate the reviewer’s helpful comments here. We apologize for our careless mistake. The word spelling“qRT-PCR”was revised at line 2 , 5, 9 and 20; the word spelling“diease”was revised at line 4 and line 13; the word spelling“colitic”was revised at line 5 and line 20; gene name “Zo-1” was revised at Table S1 and S4; the name of the species“K. marxianus, D. hansenii”was revised at Table S2.
